# Effects of Complete Oral Motor Intervention and Nonnutritive Sucking Alone on the Feeding Performance of Premature Infants: A Systematic Review and Meta-Analysis

**DOI:** 10.3390/children11010004

**Published:** 2023-12-20

**Authors:** Yu-Lin Tsai, Pei-Chun Hsieh, Ting-Yen Chen, Yu-Ching Lin

**Affiliations:** 1Department of Pediatrics, National Cheng Kung University Hospital, College of Medicine, National Cheng Kung University, Tainan 701, Taiwan; tnfsh01501@gmail.com; 2Department of Physical Medicine and Rehabilitation, National Cheng Kung University Hospital, College of Medicine, National Cheng Kung University, Tainan 701, Taiwan; peichunhsieh66@gmail.com (P.-C.H.); skywriter929@gmail.com (T.-Y.C.)

**Keywords:** oral motor intervention, nonnutritive sucking, oral motor stimulation, preterm infant, feeding performance

## Abstract

We explore the effect of complete oral motor intervention (OMI) and nonnutritive sucking (NNS) alone on oral feeding in preterm infants through a meta-analysis. We searched the Embased, Medline, Cochrane Library, and Web of Science databases for randomized controlled trials up to 8 August 2023, based on established selection criteria. Quality evaluations of the studies were carried out by applying both the Cochrane risk of bias assessment tool and the Jadad scale. The outcome measures of three clinical indicators included transition time to oral feeding, weight gain, and hospitalization duration. We conducted a meta-analysis using a random-effects model to determine the pooled effect sizes, expressed as standardized mean differences (SMDs) and their corresponding confidence intervals (CIs). Additionally, we undertook a subgroup analysis and meta-regression to investigate any potential moderating factors. Eight randomized controlled trials with 419 participants were selected. Meta-analysis revealed that receiving a complete OMI had significantly reduced transition time compared with those receiving NNS alone in preterm newborns (SMD, −1.186; 95% CI, −2.160 to −0.212, *p* = 0.017). However, complete OMI had no significant effect on shortened hospitalization duration (SMD, −0.394; 95% CI, −0.979 to 0.192, *p* = 0.188) and increased weight gain (SMD, 0.346; 95% CI, −0.147 to 0.838, *p* = 0.169) compared with NNS alone. In brief, a complete OMI should not be replaced by NNS alone. However, we were unable to draw decisive conclusions because of the limitations of our meta-analysis. Future well-designed randomized controlled trials are necessary to confirm our conclusion.

## 1. Background

The World Health Organization defines preterm newborns as those born before the gestational age (GA) of 37 weeks [1]. Global preterm [2] birth rates have not changed measurably in the previous decade, with an incidence of one in ten live births in 2020 [3]. Among 6017 moderately preterm infants across 18 neonatal intensive care units, inadequate oral feeding was the most common barrier to discharge [4]. The development of sucking behaviors in preterm infants may reflect neurobehavioral maturation and organization [5,6]. Oral alimentation requires a strong sucking effort as well as coordination between the process of swallowing, epiglottal and uvular closure of the larynx and nasal passages, and normal motility of the esophagus. This coordination is usually absent before 34 weeks of gestation [7], perhaps because the coordination between the autonomic, motoric, and behavioral subsystems is still immature at this developmental stage. Therefore, establishing a stable suck-swallow-breathe cycle in preterm infants remains challenging [8].

To improve preterm infants’ oral feeding capability, Fucile et al. [9] described an oral motor intervention (OMI) to reduce the level of oral hypersensitivity and improve the range of motion and strength of muscles used for sucking. This intervention involved stroking perioral and intraoral parts, followed by using pacifiers or fingers to induce nonnutritive sucking (NNS) before feeding. Stroking perioral and intraoral parts has been reported to enhance sucking rate and feeding efficiency [10,11]. According to Arvedson, OMI involves stimulating or moving various oral and respiratory muscles to enhance oropharyngeal function, which for preterm infants can encompass activities like NNS [12]. NNS can assist infants in achieving and maintaining physiological homeostasis and behavioral states, acquiring mature nutritive sucking patterns, and improving feeding skills [13,14]. Improper swallowing while nutritive sucking can result in complications such as aspiration pneumonia, bradycardia, hypoxia, and fatigue [15,16]. In brief, NNS can create oral feeding experiences without the added stress of fluid.

Complete OMI (OMI including NNS) and NNS alone have been suggested to improve the oral feeding performance of preterm infants. Compared with complete OMI, NNS is easy to perform, does not require professional training, and imposes a low clinical load and financial burden. However, whether a complete OMI is more effective than NNS alone remains unclear. Therefore, in this systematic review and meta-analysis of randomized clinical trials (RCTs), we compared a complete OMI and NNS alone in terms of their efficacy in improving the feeding performance of preterm infants.

## 2. Methods

The methodology for this systematic review adhered to the guidelines outlined in the Preferred Reporting Items for Systematic Reviews and Meta-Analyses (PRISMA). Additionally, we registered our review protocol on the International Platform of Registered Systematic Review and Meta-Analysis Protocols, under the registration number INPLASY2023100028.

### 2.1. Eligibility Criteria

We included RCTs that met the following PICO (population, intervention, comparison, outcome) criteria: preterm infants; complete OMI versus NNS alone; and outcomes of transition time, hospitalization duration, weight gain, feeding efficiency, and other oral feeding-related indicators. We excluded studies that did not consider transition time as an outcome, had incomplete data, lacked clarity on OMI or NNS management, and were non-RCTs. The details of the eligibility criteria as below.

#### 2.1.1. Inclusion Criteria

Studies involving preterm infants defined: born between 24 and 36 weeks of gestational age, as per the criteria specified in individual studies. Types of Studies: (1) randomized controlled trials (RCTs) focusing on the effects of nonnutritive sucking (NNS) and oral motor stimulation (OMS) on feeding outcomes in preterm infants. (2) Published full-text articles, abstracts, and conference papers where sufficient data were available. Intervention Characteristics: studies focusing on interventions involving either NNS alone or a combination of NNS and OMS, applied before or during feeds. Outcome Measures: studies must report on at least one of the following outcomes: time to transition to oral feeding, weight gain, duration of hospital stay, or feeding efficiency.

#### 2.1.2. Exclusion Criteria

Studies including infants with: (1) congenital anomalies, chronic medical complications, asphyxia, or other severe conditions that could interfere with feeding. (2) Intraventricular hemorrhage greater than grade II, necrotizing enterocolitis (NEC) stage 2 or beyond, severe perinatal asphyxia, or those requiring complex medical or surgical interventions. Types of Studies: (1) excluding case studies, reviews, editorials, commentaries, and qualitative studies. (2) Studies not written in English. (3) Studies lacking accessible full-text or adequate data for analysis.

### 2.2. Search Strategy

The PubMed, Embase, Cochrane Central Register of Controlled Trials, and Web of Science databases were searched from their inception to 8 August 2023, for studies published in English. The following key terms were used: “oral motor intervention” AND “non-nutritive suck” (see Appendix A for the full search strategy). Additionally, we manually examined the reference lists of the chosen articles to identify further relevant studies.

### 2.3. Study Selection and Data Extraction

The authors independently identified potentially relevant articles by screening titles and abstracts. Next, the full text of the selected studies was screened. The senior author, YCL, made the final decision in cases of disagreements.

Data from the chosen studies were collected using a specific data extraction form: study identification (first author and publication year), demographic characteristics (cohort age, sex, and geographic location), methodological characteristics (interventions), and outcome measures. The authors of the included articles were contacted, if necessary, to obtain clarifications for any uncertainties.

### 2.4. Quality Assessment

The quality of the selected RCTs was evaluated using the Cochrane Risk of Bias Tool 2 [17] for RCTs. Any disagreement was resolved through mutual discussion, and the senior author, YCL, made the final decision if a consensus was not reached. Review Manager version 5.3 (Cochrane, 157 London, UK) was used to visualize the risk of bias in a graph and summary table.

### 2.5. Statistical Analysis

We included all RCTs that reported the relevant outcomes in our quantitative analysis. When articles lacked standard deviation and mean data, we utilized the provided *p* values and sample sizes. The primary outcome was transition time to complete oral feeding, and the secondary outcomes were hospitalization duration, weight gain, and feeding efficiency. All outcomes are presented in terms of standardized mean differences (SMDs) and 95% CIs. The effect sizes were pooled using a random-effects model. We also conducted a random-effects meta-regression to investigate whether the primary outcome varied depending on the study characteristics (continuous variables), including GA and birth weight. For the categorical variables, such as transition time, the included trials were grouped first, then the summarized effect sizes of the subgroups were calculated separately. Nonoverlapping 95% CIs indicated significant differences between the subgroups. The *I*^2^ statistic was used to evaluate between-study heterogeneity, which was regarded as significant at >50%. We used funnel plots and the Egger test to assess publication bias, and a 2-tailed *p* value of <0.1 indicated statistical significance. We conducted a sensitivity analysis on the primary outcome by removing one trial at a time and analyzing the remaining trials to investigate whether the effect resulted from a single study. The Comprehensive Meta-Analysis Software version 3 (Biostat, Englewood, NJ, USA) was used for analysis.

### 2.6. Certainty of Evidence

Evidence quality for the main outcome was evaluated using the Grading of Recommendations Assessment, Development and Evaluation approach, generally finding high certainty due to the inclusion of only RCTs. The final assessment considered factors like bias risk, precision, consistency, relevance, and publication bias.

## 3. Results

### 3.1. Study Selection

We identified 451 articles in the initial search; of these, 8 RCTs met our inclusion criteria (Figure 1) [18,19,20,21,22,23,24,25]. Table 1 presents the characteristics of these 8 RCTs.

### 3.2. Risk-of-Bias Assessment

Among the included RCTs, 4 had a high/unclear risk of allocation concealment, and 5 had a high/unclear risk of participants or personnel blinding because of the nature of the intervention. The blinding of outcome assessors was unclear in 4 studies (Figure 2). Most studies reported all outcomes, and the participants were followed up until discharged. All RCTs scored >5 on the modified Jadad scale (Table 2).

### 3.3. Outcome Measures

All 8 RCTs (a total of 848 participants) analyzed the primary outcome. The mean GA ranged from 30 to 33 weeks. Meta-regression analyses were conducted using the mean GA and birth weight. Newborns with complications such as intraventricular hemorrhage, bronchopulmonary dysplasia, severe perinatal asphyxia, severe sepsis, and congenital disease were excluded from the analyses.

The meta-analysis was conducted for 4 frequently used outcome indicators: transition time, hospitalization duration, weight gain, and feeding efficiency. Transition time was defined as the interval days between intervention and independent oral feeding. Hospitalization duration was the days between admission and discharge. Weight gain referred to the weight gained between hospital admission and discharge. Feeding efficiency was analyzed in terms of feeding volume, duration, or rate.

### 3.4. Transition Time

All included RCTs reported the transition time. Significant heterogeneity (*I*^2^ = 94.7%) was noted among the studies. Thus, a random-effects model was used to calculate the mean effect size. The meta-analysis revealed that a complete OMI significantly shortened the transition time compared with the effect of NNS alone (SMD, −1.186; 95% CI, −2.160 to −0.212, *p* = 0.017; Figure 3). The meta-regression revealed no influence on effect size considering GA (β = −0.4350, *p* = 0.22) or birth weight (β = 0.0014, *p* = 0.46).

### 3.5. Hospitalization Duration

Six trials (299 participants) analyzed the duration of hospitalization. A random-effects model was used because of significant heterogeneity (*I*^2^ = 82.7%). The duration of hospitalization was not shorter in the complete OMI group than in the NNS alone group (SMD, −0.394; 95% CI, −0.979 to 0.192, *p* = 0.188; Figure 4).

### 3.6. Weight Gain

Seven studies (373 participants) recorded information on weight gain. A random-effects model was used because of significant heterogeneity (*I*^2^ = 80.8%). The amount of weight gain in the OMI group was not greater than that in the NNS group (SMD, 0.346; 95% CI, −0.147 to 0.838, *p* = 0.169; Figure 5).

### 3.7. Complication

Three studies noted that during the intervention period, no adverse events such as bradycardia, apnea, desaturation, aspiration, or hypothermia occurred that required discontinuation of the interventions [19,21,24]. Two studies reported that episodes of apnea, bradycardia, or oxygen desaturation during the oral feeding session were recorded. However, there is no specific mention of the frequency of these complications or if there was any significant difference between the intervention and control groups regarding these episodes [22,25]. One study did not specifically mention complications such as apnea, desaturation, or aspiration within this document [20]. One study recorded the incidence of adverse reactions, which included apnea, abdominal distension, and decreased blood oxygen saturation, was statistically significantly lower in the intervention group compared to the control group. There was no significant difference in the amplitude of desaturations between the two groups. This suggests that the intervention may have had a beneficial effect in reducing the incidence of certain feeding-related complications. However, the small sample size and the limited number of observed complications suggest that more robust and larger-scale studies are needed to conclusively determine the intervention’s effectiveness [23]. One study, which included the most premature preterm infants in the eight RCTs, during the application of the protocol, 13 (54.17%) premature babies exhibited mild bradycardia or desaturation. Of these, 11 babies (84.62%) were able to continue with the protocol after a brief pause, while 2 (15.38%) had to have the protocol interrupted and restarted in the following intervention session. Despite these incidents, the intervention was completed for all cases mentioned. The study also discusses the potential complications associated with prolonged tube feeding, such as damage to the gastrointestinal mucosa and increased incidence of apnea and bradycardia due to vagal stimulation, indicating that reducing tube feeding time could minimize such risks [18].

With the analysis of all eight articles, it is clear that while some studies reported instances of mild bradycardia or desaturation, these complications were generally brief and managed within the study protocols without requiring discontinuation of the intervention. The studies focused on the positive outcomes of the interventions on reducing the time to achieve full oral feeding and did not report significant feeding-related complications, suggesting that the interventions were generally safe and well-tolerated.

### 3.8. Certainty of Evidence

The certainty of evidence for the OMI-mediated reduction in transition time was moderate; the reason for downgrade was the large 95% CI (Table 3).

## 4. Discussion

In our systematic review and meta-analysis, we revealed that compared with NNS alone, complete OMI improved the oral feeding performance of preterm infants, significantly shortening the time to independent oral feeding as evaluated by transition time. However, weight gain during the hospital stay and hospitalization duration were similar between the two groups.

Few meta-analyses or reviews have compared transition time between complete OMI and NNS, precluding adequate comparisons with our data. A meta-analysis indicated that complete OMI can significantly shorten transition time [26]. A complete OMI promotes the maturation of oral motor skills to prevent oral feeding difficulties [9], such as disorganized sucking patterns, oral hypersensitivity, and absent suck-swallow-breathe coordination. NNS was reported to enhance feeding performance [27,28,29], specifically the coordination of jaw and tongue movements, thereby accelerating the acquisition of mature nutritive sucking patterns [14,25,30]. We speculate that complete OMI and NNS alone exerted different effects on transition time because NNS does not improve the coordination between swallowing and breathing. For example, respiratory patterns do not change during NNS as they do during nutritive sucking [2]. In brief, complete OMI comprehensively improves the suck-swallow-breathe coordination, accelerating the maturation of feeding skills and shortening the time to independent oral feeding.

Our meta-analysis found that although four of seven trials indicated a slightly shortened hospitalization duration, this parameter did not significantly differ between complete OMI and NNS only. A meta-analysis also revealed that OMI did not significantly reduce the duration of hospitalization [26]. Although a shorter transition time may theoretically imply a shorter hospitalization duration because of the reduced complications in immature nutritive sucking skills, such as choking, this may be confounded by the fact that hospitals do not have a strong discharge standard. Another confounder can be the definition of hospitalization duration. We calculated it from admission to discharge; however, data pertaining to the period between intervention and discharge may better reflect the efficacy of an intervention. Only two of the included RCTs recorded the time from intervention to discharge, and they reported a longer time to discharge for the NNS group. One of the aforementioned trials revealed that the outcome was the opposite when hospitalization duration was calculated from admission.

Our data indicated that weight gain was comparable between the two groups. We believe that the rate of weight gain based on their own weight (g/kg/day) is a better index than pure weight gain rate (g/day); however, because of the different indices in the included studies, we could only compare weight gain between seven trials. Of the included RCTs, only one [24] reported the rate of weight gain from the day of intervention to the day of achieving independent oral feeding and concluded that this rate was significantly higher in the OMI group than in the NNS group. As mentioned, few reviews or RCTs had performed the same comparisons as our study. Another study measuring the rate of weight gain from admission to discharge showed comparable effects between complete OMI and routine care; however, the authors did not mention whether routine care included NNS [31]. Gavage feedings require less energy for the feeding process, and excessive oral feedings may tire an infant and reduce the weight gain rate [32].

This review and meta-analysis have several strengths. First, it is the first meta-analysis comparing the effect between complete OMI and NNS alone. Second, we prove that complete OMI is better than NNS alone, which means a professional technique of OMI should not be replaced by a simple NNS alone. Third, in preterm babies, shortening the transition time to complete oral feeding not only enhances feeding performance and reduces financial pressure, but also alleviates the worries of their parents who see their babies with a tube in their tiny mouths. We believe this is the most important aspect.

Our meta-analysis has some limitations. First, the OMI varied slightly across the included RCTs; five trials designed the program following that designed by Fucile [8], whereas the other three studies described the details of their program similar to that proposed by Fucile et al. but that differ in times and duration. Second, some studies did not provide the details of NNS such as the frequency and program; nonetheless, we excluded the study that mentioned only pacifier use. Moreover, the frequency of NNS differed between the included RCTs. Third, outcome measures and indices varied across the included studies; therefore, we could only compare the OMI and NNS groups in terms of transition time, hospitalization duration, and weight gain. We did not assess other relevant outcomes, such as the Preterm Oral Feeding Readiness Assessment Scale scores [19,23], sucking power [22], or volume and rate of milk intake [19,22,23]. Finally, the included studies tended to exclude preterm newborns with complications or respiratory, cardiovascular, neurological, and digestive disorders; however, these newborns constitute an important proportion of preterm infants with pediatric feeding disorders. The aforementioned tendency limited the generalizability of our findings. Nevertheless, our findings support the notion that complete OMI cannot be replaced by NNS alone, and these OMI effectively shorten the time to achieving independent oral feeding.

## 5. Conclusions

Preterm newborns receiving a complete OMI had significantly reduced transition time compared with those receiving NNS alone. However, whether complete OMI can also shorten hospitalization duration, increase weight gain, and improve other feeding performance remains unclear. Future well-designed, large-scale RCTs, particularly those including various outcome indicators, are warranted to validate and extend our findings.

## Figures and Tables

**Figure 1 children-11-00004-f001:**
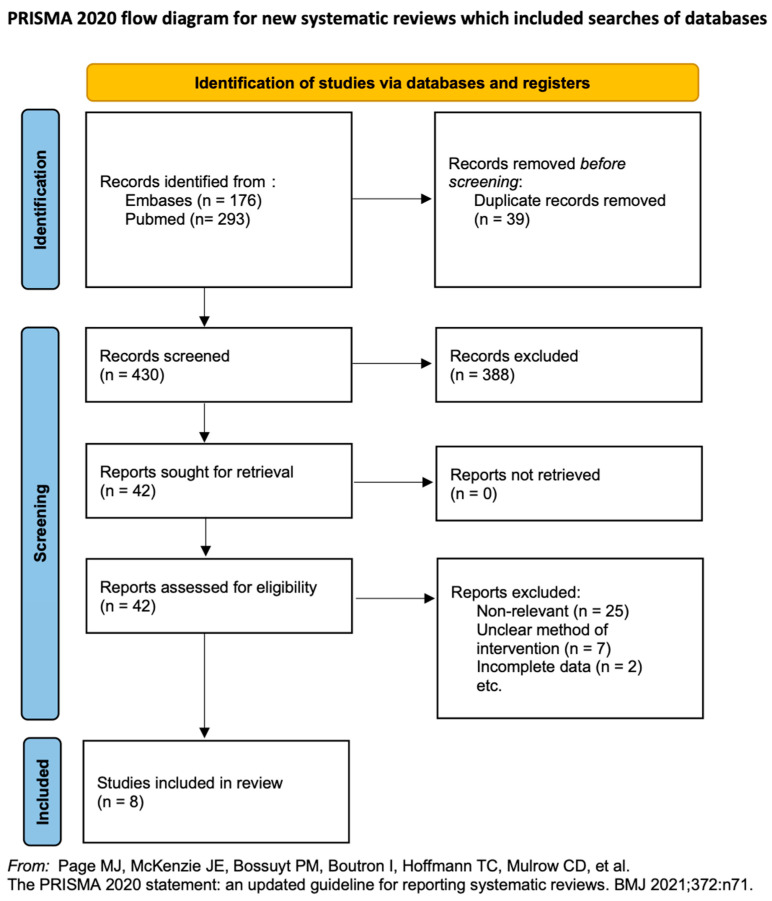
Literature screening process and results [18,19,20,21,22,23,24,25].

**Figure 2 children-11-00004-f002:**
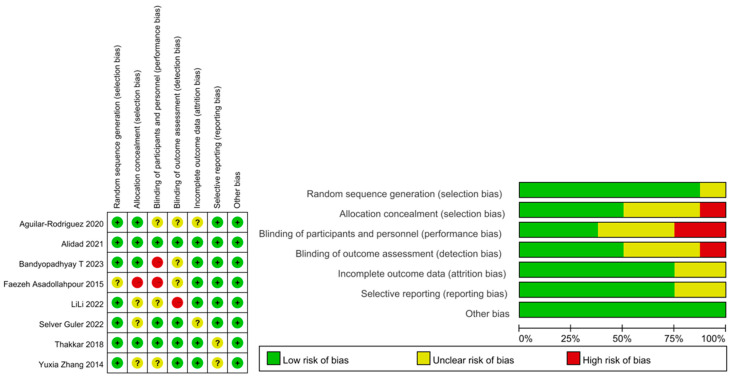
Summary graph and table for risk of bias [18,19,20,21,22,23,24,25].

**Figure 3 children-11-00004-f003:**
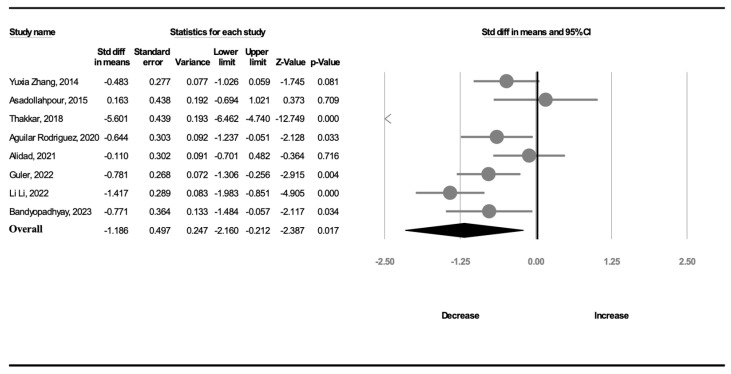
Forest plot of standardized mean differences in transition time [18,19,20,21,22,23,24,25].

**Figure 4 children-11-00004-f004:**
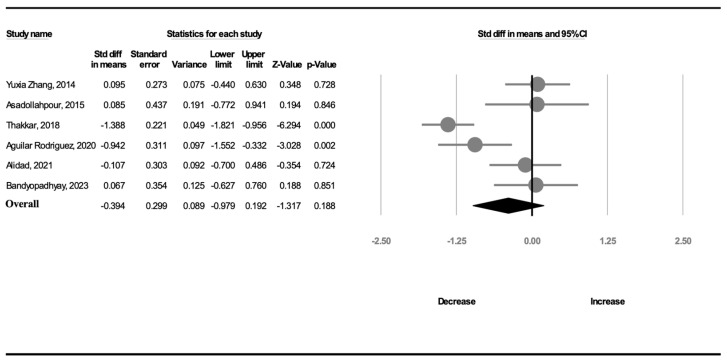
Forest plot of standardized mean differences in hospitalization duration [18,19,20,21,24,25].

**Figure 5 children-11-00004-f005:**
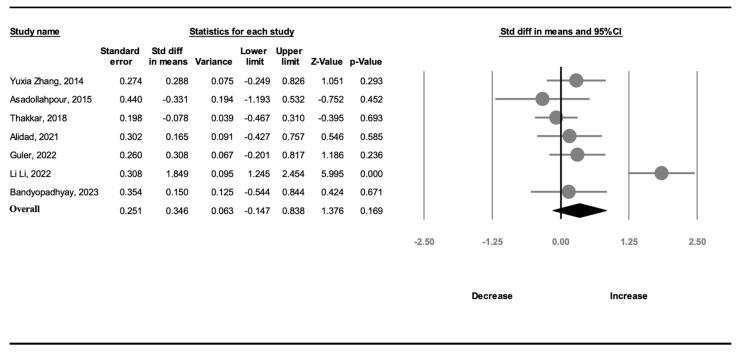
Forest plot of standardized mean differences in weight gain [19,20,21,22,23,24,25].

**Table 1 children-11-00004-t001:** Characteristics of all included research.

			No. of Participants		Postmenstrual Age (Mean ± SD, Weeks)		Birth Weight (Mean ± SD, Grams)		
Research	Country	Study Design	Complete OMI (OMI with NNS) (M/F)	NNS Only (M/F)	Complete OMI (OMI with NNS)	NNS Only	Complete OMI (OMI with NNS)	NNS Only	Outcome Measures
Aguilar-Rodríguez et al. [18]	Spain	RCT	13/11	14/8	28.36 ± 1.37	28.05 ± 1.31	1110.42 ± 252.40	1162.05 ± 217.08	Transition time, hospital duration, days from beginning until sucked the FOI30% in first 5 min and FOI100%
Alidad et al. [19]	Iran	RCT	15/15	9/21	32.4 ± 2.3	32.6 ± 1.7	2026.8 ± 627.1	2158.1 ± 694.4	Transition time, hospital duration, weight gain, POFRAS, volume intake of milk
Asadollahpour et al. [20]	Iran	RCT	5/5	6/5	30.01 ± 1.76	30.18 ± 1.77	1343.01	1406.36	Transition time, hospital duration, weight gain
Bandyopadhyay et al. [21]	India	RCT	12/4	12/4	29.93 ± 1.65	29.90 ± 1.64	1269.25 ± 276.5	1348.0 ± 300.67	Transition time, hospital duration, weight gain, episode of complications
Guler et al. [22]	Turkey	RCT	16/14	15/15	26–30	26–30	1267.0 ± 276.6	1266.7 ± 233.6	Transition time, hospital duration, weight gain, sucking capacity, sucking power, head circumference
Li et al. [23]	China	RCT	17/13	16/14	32.36 ± 1.45	32.06 ± 1.53	1730.0 ± 560	1650.0 ± 440	Transition time, weight gain, POFRAS, sucking amount and rate, adverse reactions
Thakkar et al. [24]	India	RCT	28/23	24/27	32.1 ± 0.8	32.29 ± 0.6	1314.04 ± 105	1316.13 ± 80	Transition time, hospital duration, weight gain, volume intake of milk, rate of intake
Zhang et al. [25]	China	RCT	15/14	11/14	31.0 ± 1.4	30.9 ± 1.7	1579.3 ± 280.7	1548.2 ± 233.8	Transition time, hospital duration, weight gain, volume intake in first 5 min

OMI = oral motor intervention; NNS = nonnutritive sucking; M = male; F = female; POFRAS = preterm infant oral feeding readiness assessment scale.

**Table 2 children-11-00004-t002:** Risk of bias assessed by Jadad scale.

	Aguilar-Rodríguez et al. [18]	Alidad et al. [19]	Asadollahpour et al. [20]	Bandyopadhyay et al. [21]	Guler et al. [22]	Li et al. [23]	Thakkar et al. [24]	Zhang et al. [25]
Described as randomized *	1	1	1	1	1	1	1	1
Appropriate randomization method **	1	1	0	1	1	1	1	1
Described as blinding ***	0.5	0.5	0.5	0.5	1	0	1	0.5
Appropriate blinding method **	1	1	1	1	1	0	1	1
Description of withdrawals and dropouts *	1	1	0	1	0	0	1	1
Description of inclusion/exclusion criteria *	1	1	1	1	1	1	1	1
Description of method used to assess adverse effects *	1	0	1	1	0	1	1	1
Description of statistical analysis methods *	1	1	1	1	1	1	1	1
Score	7.5	6.5	5.5	7.5	6	5	8	7.5

* A study scores 1 for “yes” and 0 for “no”; ** A study scores 0 if no description is given, 1 if the method is both described and appropriate, and if the method is described but inappropriate; *** A study scores 1 for “double-blinded”, 0.5 for “single-blinded” and 0 for “no”.

**Table 3 children-11-00004-t003:** Certainty of evidence.

Quality Assessment						Summary of Findings, SMD (95% CI)	
Number of Participants (Studies), Follow-Up Period	Risk of Bias	Inconsistency	Indirectness	Imprecision	Publication Bias	Transition Time	Certainty of Evidence
848 (8), during admission	No serious limitation ^a^	Serious limitation ^b^	No serious limitation ^c^	No serious limitation ^d^	No serious limitation ^d^	−1.186 (−2.160, −0.212) ^e^	Moderate ⨁⨁⨁◯

CI: confidence interval; SMD: standardized mean difference. ^a^ Most studies included scored low risk of bias during assessment. ^b^ I^2^ score was above 50%. ^c^ No indirectness was detected in this outcome. ^d^ The upper and lower limit of 95% CI. ^e^ This was calculated by pooling the included 8 articles.

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
