# Peer review of "Effects of Complete Oral Motor Intervention and Nonnutritive Sucking Alone on the Feeding Performance of Premature Infants: A Systematic Review and Meta-Analysis"

_children, 2023, doi:10.3390/children11010004_

Round 1

Reviewer 1 Report

Comments and Suggestions for Authors

Complete oral motor interventionOMIand nonnutritive suckingNNSare important tools for feeding interventions in preterm infants and have been suggested to improve the oral feeding performance of preterm infants. The study compared a complete OMI and NNS alone in terms of their efficacy in improving the feeding performance of preterm infants through the systematic review and meta-analysis of randomized clinical trials. Althought there is no similar Meta-analysis to summarize the effect of these two interventions on feeding performance of premature infants, there are still some problems in this article. Here are a few comments:

1、  The search time mentioned in the Abstract is August 2023, while the search time in the Search Strategy is August 8, 2023.

2、  The inclusion and exclusion criteria of the articles are not clear enough, and the inclusion can be shown in sections, such as the type of research, the type of articles and so on the exclusion criteria are the same as above.

3、  Did the article conduct an artificial search for references to relevant reviews?

4、  Only the transition time to oral feeding, weight gain, and hospitalization duration were included in the article, whether the effects of these two approaches on feeding-related complications need to be included in the analysis

5、  The forest plot presented in the article is not standardized enough, you can show the X-axis of the forest plot and the invalid line with solid lines, the remaining parallel to the Y-axis scale line can not appear.

Comments on the Quality of English Language

Minor editing of English language required

Reviewer 2 Report

Comments and Suggestions for Authors

Very interesting, orijinal and well planned study. Please kindly find the reviewer report attached.
